# *Serendipita indica*—A Review from Agricultural Point of View

**DOI:** 10.3390/plants11243417

**Published:** 2022-12-07

**Authors:** Sana Saleem, Agnieszka Sekara, Robert Pokluda

**Affiliations:** 1Department of Vegetable Sciences and Floriculture, Faculty of Horticulture, Mendel University in Brno, Valticka 337, 691 44 Lednice, Czech Republic; 2Department of Horticulture, Faculty of Biotechnology and Horticulture, University of Agriculture, 31-120 Krakow, Poland

**Keywords:** beneficial microorganism, endophytic fungus, symbiosis, stress tolerance, biofertiliser, plant growth, quality, gene regulation, phytohormones, nanoparticle

## Abstract

Fulfilling the food demand of a fast-growing population is a global concern, resulting in increased dependence of the agricultural sector on various chemical formulations for enhancing crop production. This leads to an overuse of chemicals, which is not only harmful to human and animal health, but also to the environment and the global economy. Environmental safety and sustainable production are major responsibilities of the agricultural sector, which is inherently linked to the conservation of the biodiversity, the economy, and human and animal health. Scientists, therefore, across the globe are seeking to develop eco-friendly and cost-effective strategies to mitigate these issues by putting more emphasis on the use of beneficial microorganisms. Here, we review the literature on *Serendipita indica*, a beneficial endophytic fungus, to bring to the fore its properties of cultivation, the ability to enhance plant growth, improve the quality of produced crops, mitigate various plant stresses, as well as protect the environment. The major points in this review are as follows: (1) Although various plant growth promoting microorganisms are available, the distinguishing character of *S. indica* being axenically cultivable with a wide range of hosts makes it more interesting for research. (2) *S. indica* has numerous functions, ranging from promoting plant growth and quality to alleviating abiotic and biotic stresses, suggesting the use of this fungus as a biofertiliser. It also improves the soil quality by limiting the movement of heavy metals in the soil, thus, protecting the environment. (3) *S. indica*’s modes of action are due to interactions with phytohormones, metabolites, photosynthates, and gene regulation, in addition to enhancing nutrient and water absorption. (4) Combined application of *S. indica* and nanoparticles showed synergistic promotion in crop growth, but the beneficial effects of these interactions require further investigation. This review concluded that *S. indica* has a great potential to be used as a plant growth promoter or biofertiliser, ensuring sustainable crop production and a healthy environment.

## 1. Introduction

A diverse range of microorganisms in soil play critical functions, such as nutrient acquisition, organic matter cycling, soil and plant health maintenance, soil restoration and ecosystem primary production, and are thus considered as beneficial organisms [1]. By increasing crop yield, quality, and shielding the plants from various biotic and abiotic challenges, these microbes have shown several positive effects on the farming system. In addition, environmental protection, an increasing concern across the globe, is the main advantage these creatures provide [2]. Moreover, the rising demand and growing awareness of high-quality food has stimulated the development of sustainable and environmentally friendly agricultural production practices. These methods could be achieved by reducing the usage of chemicals and encouraging the application of beneficial microorganisms [3]. Apart from contributing to sustainability, beneficial microorganisms are also economically efficient [4]. These microorganisms can be symbiotic or non-symbiotic bacteria, actinomycetes, and mycorrhizal and endophytic fungi [5]. Throughout the study of various microorganisms, an emphasis is placed on studying endophytic fungi, as they can yield tangible beneficial effects within the host plant, such as improve plant growth, quality, and enhance host resistance to abiotic and biotic stresses, thereby confirming their significance in the agricultural sector [6,7].

*Serendipita indica*, formerly known as *Piriformospora indica*, belonging to the order *Sebacinales* (*Basidiomycota*), is one of the beneficial endophytic fungi known to possess numerous advantages and has been studied extensively for decades. *S. indica*, obtained by Ajeet Verma from the roots of *Prosopis juliflora* and *Zizyphus nummularia* in the Thar desert in Northwest India, is characterised by the formation of pear-shaped spores known as chlamydospores [8]. These spores are produced by thin walled, hyaline, and septate hyphae [9]. *S. indica* is distinguished by the unique trait that it can be cultivated without any plant material on a variety of artificial media, such as aspergillus medium, modified aspergillus medium, potato dextrose agar (PDA) or broth (PDB), malt extract, modified Melin-Norkrans (MMN) medium [10], and jaggery containing medium [11]. This property differentiates it from arbuscular mycorrhizal fungi (AMF), with which it shares biological similarities (Figure 1).

This endophytic fungus can colonise a wide range of plant species and develop symbiotic associations with them [12,13]. Mainly, colonisation takes place in the root zone and begins with the germination of chlamydospores, followed by the formation of a hyphal network on and inside the root. Hyphae form branches and continue to grow by penetrating the subepidermal layers of roots, and eventually, cover the rhizodermal and cortical cells (Figure 2). Mature root segments show intra- and intercellular colonisation patterns, while conductive tissues are free from colonisation. This fungus colonises different root zones, with maximum colonisation in the zone of cell differentiation. In addition, the fungus penetrates root hair cells, forming hyphae from germinating spores [14,15] (Figure 3). This process begins with the biotrophic phase, followed by the cell death phase, to build the symbiotic relationship with the host rhizosphere [16,17].

Moreover, a successful symbiosis between *S. indica* and the host plant depends on administering the right quantity of inoculum, which can be assessed using a qPCR test [18]. According to Rokni et al. [19], optimising the *S. indica* inoculum concentration had an impact on the host plant’s development. Plant growth was enhanced by a 1–3% *w*/*w* concentration; however, higher doses did not show any positive benefits.

The production of calcium and lectin protein kinase induced during symbiotic association are suggested to be the early signalling factors in *S. indica*–host plant colonisation [20,21]. Additionally, the synthesis of the plant hormone ethylene is considered crucial for the interaction between plant hosts and fungi [22].

*S. indica* has been reported to provide numerous advantages to host plants upon colonisation of roots. These advantages include enhanced plant growth, nutrient uptake, and antioxidant activities; increased photosynthetic pigments, crop quality and yields; as well as the ability to mitigate various biotic and abiotic stresses [3,23,24,25,26,27,28]. Although mycelia and spores from fungi are typically considered the beneficial form of inoculum, some studies have demonstrated that cell wall extracts and culture filtrates from *S. indica* have a positive effect on plant growth [29,30].

Furthermore, to commercialise *S. indica*, biofertiliser formulation, known as ‘rootonic’, has been developed by culturing this fungus in a bioreactor [12,31,32] and mixing with carrier magnesium sulphite [33]. This endophytic fungus is proving to be an efficient source of bio-inoculant for the agricultural sector, with the least environmental hazards and improved agricultural sustainability.

In this review, we have summarised the effect of *S. indica* on the host plant and the underlying mechanisms. This review also includes the management of various plant stresses by *S. indica* and a brief description of its interactions with nanoparticles.

## 2. *S. indica* as a Growth Promoter

*S. indica* has been found to be a prime beneficial microorganism that improved the growth and development of various plant species under normal and stress conditions [3,28] (Figure 4). This endophytic fungus improved the germination rate in various plant species, such as chilli, cabbage, cucumber, eggplant, maize, okra, spinach, rice, and tobacco [11,34,35,36]. Improved seed germination under intense cold conditions in beetroot, carrot, cauliflower, onion, and radish has also been reported [17]. The application of *S. indica* has been reported to promote seedling growth and development by enhancing root and shoot growth, biomass, photosynthetic pigment production, and seedling vigour. [36,37,38]. Sheramati et al. [39] reported that *S. indica* improved the growth of *Arabidopsis* and tobacco seedlings by enhancing the nitrogen accumulation and expression of genes governing nitrogen reductase and the starch degrading enzyme glucan-water dikinase in their roots. These enzymes are responsible for nitrogen and starch metabolism in seedlings, which is essential for their growth and development.

This fungus is reported to promote the plant vegetative development in terms of height, leaf number, shoot and root growth, fresh weight, dry weight, as well as photosynthetic pigment and phytohormone synthesis [3,22,25,34,40,41]. Stimulation of nutrient uptake and their efficiency is also reported due to *S. indica* colonisation leading to improved plant growth, quality, and yield [25,42]. Improvement in other plant physiological attributes, such as inflorescence development, duration of flowering, flower size and number of flowers, have been reported with the application of the endophytic fungus *S. indica* [25,30,34,43]. Therefore, *S. indica* promotes plant growth and development by inducing earliness of host plant reproduction [44]. Furthermore, *S. indica* improved crop yields by elevating chlorophyll content, flower setting, grain yield, and pod number and size in crops, such as black pepper [43], fennel [45], rice [44,46], rapeseed [25], sweet potato [47], sunflower [48], and tomato [49,50].

In addition to improving the growth and production of crops, *S. indica* has improved the quality of the products. This impact may be attributable to increased root growth and root hair formation in plants treated with *S. indica*, which results in improved mineral nutrition acquisition and water uptake [9,45,51]. Singhal et al. [24] further proposed that nutrient uptake promoted by this endophytic fungus could be attributable to fungal hyphae penetrating deeper into the soil than root hair and thus absorbing more nutrients from the soil. Assimilation of macronutrients, such as nitrogen, phosphorus, potassium, sulphur, and magnesium is improved in crops, such as *Poncirus trifoliata, Triticum aestivum, Brassica napus, Oryza sativa, Panicum miliaceum, Arabidopsis thaliana,* etc. [7,25,46,52,53,54].

*S. indica* improves the availability of nitrogen by transforming the unavailable form into nitrate, which is taken up by plants [9]. Moreover, it stimulates the expression of the gene governing NADH-dependent nitrate reductase, which is responsible for nitrate assimilation in crops [9,55,56]. *S. indica* is able to convert organic phosphorus into plant-available form, thus improving the phosphorus uptake by plants under normal as well as under phosphorus-deficient conditions [17,57,58]. It also improves the acid phosphatase and alkaline phosphatase activities, leading to an improvement in phosphorus uptake in rice and rapeseed [58,59]. In addition to promoting soil phosphatase activities, *S. indica* upregulated the expression of the phosphate transporter genes (*PT3, PT5, and PT6*) in *Poncirus trifoliata* and thereby enhanced the phosphorus absorption by the plant [7]. *S. indica* colonisation improved the potassium concentration in tomatoes, resulting in higher levels of lycopene and ascorbic acid and therefore improving the quality of tomatoes [60]. Several studies have reported a significant increase in absorption of other macronutrients, such as calcium, magnesium, and sulphur [7,25,61,62,63]. An increase in the absorption of magnesium and sulphur is suggested due to the enhanced expression of transporters *Pi*MgT1 and *Pi*SulT, respectively, in plants treated with *S. indica* [62,63]. Moreover, the application of *S. indica* improved the acquisition of iron and zinc in rapeseed [25] and zinc in lettuce and *Brassica napus* [64,65]. Furthermore, inoculation with *S. indica* increased iron content in sugarcane plants by enhancing iron transportation [33,66]. Therefore, the application of *S. indica* has proven to enhance nutrient uptake in plants, and thus is considered as an emerging plant growth promoter. Table 1 summarises the effects of *S. indica* on the growth and quality of different host plants.

Furthermore, *S. indica* not only improves the nutritional status of plants, but also develops resilience in host plants against many stresses. Additionally, it reduces the concentration and effect of various harmful substances and heavy metals, thus enhancing soil health [91], which is further discussed in the review.

## 3. The Mechanism behind the Improvement in Growth, Yield, and Quality of Crops by *S. indica*

Since endophytic fungi are known to enter and survive inside a host plant, these fungi need to pass through the host’s defence system, which comprises specific receptors, such as pattern recognition receptors and intracellular receptors. These receptors capture the microbe or pathogen by recognizing the microbe- or pathogen-associated molecular patterns (MAMPs or PAMPs) located on the cell surface, thus resulting in MAMP- or PAMP-triggered immunity in the hosts. Other defence-related receptors are intracellular receptors, which recognise effector proteins secreted by the microbes, leading to effector-triggered immunity (ETI) [92,93]. Endophytic fungi produce some molecular patterns, such as β-glucan and chitin, which are recognised by the pattern recognition receptors present on host cell walls [94]. Then, chitin is converted into chitin oligomers by chitinase produced by the host plant, resulting in defensive reactions [95,96]. However, fungal endophytes produce chitin deacetylases that convert chitin oligomers to deacetylated chitosan oligomers, which are thus not recognised by the host plant’s receptors and prevent triggering the plant immune system [93]. In the case of *S. indica*, β-glucan-stimulated immunity is suppressed due to the presence of the fungal glucan-binding 1 protein gene, which encodes β-glucan-binding lectin that modifies the fungal cell wall, thus suppressing the recognition process in plant hosts [97]. Therefore, endophytic fungi have the capacity to survive in the plant root and offer various benefits, such as improving plant immunity and enhancing the response of plants toward various biotic and abiotic stresses [98] in exchange for carbohydrates as food.

Once the fungus successfully enters the host plant, it begins to develop a symbiotic association with the plant. It exhibits an impact on the plant by inducing an array of physiological and molecular changes, such as enhancing transportation and uptake of nutrients, promoting nutrient solubilisation, reducing heavy metal toxicity, and maintaining optimum pH. These modifications constitute the direct mechanism of *S. indica* mediated plant growth [56,91]. In contrast, indirect mechanisms are affiliated with changes in hosts, such as enhanced antioxidants and secondary metabolites production [99,100], decreased anti-nutrient contents, as well as modulation of gene expression and phytohormone production [91]. Moreover, *S. indica* enhances chlorophyll content, thus improving the rate of photosynthesis, and consequently, the growth of the plant [3]. Hua et al. [100] performed a global metabolic analysis in Chinese cabbage inoculated with *S. indica* and reported improvements in metabolite production, fatty acid content, and activity of the TCA cycle, leading to increased carbohydrate production in plants to support the growth and development of the root system.

Furthermore, *S. indica*-mediated growth involves interactions with phytohormones, such as auxin, gibberellin, cytokinin, and ethylene [101]. This fungus interferes with the synthesis and signalling of phytohormones to stimulate plant growth, development, and defence responses. Plants respond to *S. indica* by adjusting their hormone levels in the roots to ensure the growth and colonisation of this fungus [101]. Auxin, the major growth phytohormone, is reported to promote the growth and development of host plants under *S. indica* colonisation by regulating auxin-related genes and proteins, which are involved in transport and signal transduction of auxin, such as auxin influx carriers (*AUX1*)*,* auxin efflux carriers (*PINs*), and auxin receptor (*TIR1*) [102]. Moreover, Dong et al. [103] reported the upregulation of auxin-responsive genes, which resulted in improving plant growth. In Chinese cabbage, auxin and its intermediates were enhanced under *S. indica* colonisation [100]. Other phytohormone levels, including gibberellin and ethylene, are also enhanced in *S. indica*-colonised plants, resulting in earlier blooming and better root colonisation [91,104,105]. Enhanced expressions of gibberellin biosynthetic genes, such as gibberellin 20-oxidase 2, gibberellin 3-oxidase 1, and gibberellin requiring 1 was reported in response to *S. indica* application [101]. Additionally, the cytokinin levels were elevated in plants upon *S. indica* colonisation by upregulating cytokinin receptor genes (*CRE1*, *AHK2,* and *AHK3*), cytokinin-responsive gene (*ARR*), and trans-zeatin cytokinin biosynthesis genes, thus enhancing plant growth [106,107].

Gene regulation is suggested as an important part of plant growth promotion mediated by *S. indica* colonisation. As mentioned, several gene expressions are regulated by *S. indica* colonisation to ameliorate the phytohormonal balance in the host plant. According to Ghorbani et al. [108] and Yang et al. [7], the use of *S. indica* enhanced the expression of host genes responsible for the uptake of nutrients, such as iron (*IRO2, YSL1,* and *FRDL1*) and phosphorus (*PtPT3*, *PtPT5,* and *PtPT6*) in rice and *Poncirus trifoliata*. Moreover, it has affected the expression of various other genes in host plants, such as sucrose synthase (*SUS*) and invertase (*INV*) genes, to improve the phosphorus concentration in *Arabidopsis thaliana* by altering the sugar metabolism and obtaining fructose and glucose from the host plant [109]. Furthermore, *S. indica* resulted in the upregulation of phytochelatin biosynthesis genes, such as *PCS1* and *PCS2* in rice plants, enhancing the phytochelatin that removes the toxic metals from the host plants [108]. In Brassica napus, expressions of genes, such as trans-2, 3-enoyl-CoA reductase (*BnECR*) and 3-ketoacyl-CoA synthase (*BnFAE1)* were downregulated by applying *S. indica*. These genes are responsible for encoding some enzymes that cause the biosynthesis of erucic acids, considered as an unhealthy substance [25]. Moreover, rapeseed fatty acid content was enhanced by upregulating the expression of fatty acid regulation genes, such as 3-ketoacyl-CoA reductase 1 (*Bnkcr1*) and 3-ketoacyl-CoA reductase 2 (*Bnkcr2*) [25]. Furthermore, the application of *S. indica* to tobacco seedlings increased the expression of many genes involved in plant metabolism, including nitropropane dioxygenase (*2NpdO*), glucan-water dikinase (*GWD*), and nitrate reductase (*Nia2*) [39] (Figure 5).

Therefore, *S. indica* has an impact on a variety of plant physiological and biochemical processes, which are modulated at the molecular level through genes regulation. The integrated changes in plant physiological, biochemical, and molecular processes that are affected by the endophytic fungus is responsible for enhancing plant growth and development.

## 4. Influence of *S. indica* on Biotic and Abiotic Stress

Plants are exposed to various forms of stressful conditions that limit their growth, development, and production, due to the formation of reactive oxygen species (ROS), e.g., hydroxyl (OH·^−^), superoxide (O_2_·^−^), and hydrogen peroxide (H_2_O_2_). These stresses are classified as biotic and abiotic stresses. Biotic stresses include pathogen infection, whereas abiotic stresses include drought, salinity, heat, and cold stress [110]. Nevertheless, plants are bestowed with various protective barriers against these stresses, such as the production of secondary metabolites, phytohormones, toxic substances that halt the enzymatic activity of pathogens, and some physical barriers, such as cell walls and cuticles, to limit pathogen entry [9]. Moreover, plants produce various solutes, such as proline, sucrose, polyols, glycine, and betaine, as well as antioxidative enzymes, such as catalase, ascorbate peroxidase, and superoxide dismutase as a protection against various stresses. However, ROS production can be higher under various stresses than the plant’s defence system, leading to oxidative stresses [111]. Therefore, controlling these stresses is essential to preventing further crop losses.

*S. indica* has been reported to alleviate both biotic and abiotic stresses upon colonisation, protecting the host plants from the adverse effects of stressors [14].

### 4.1. Biotic Stress Management by S. indica

*S. indica,* as one of the most promising microorganisms, can protect plants against various stresses by regulating multiple processes, such as the synthesis of antioxidants, secondary metabolites, osmolytes, defence-related phytohormones, etc. [9]. Resistance to biotic stresses, such as bacterial, viral, and fungal diseases in plant hosts have been reported following *S. indica* colonisation [9,12,112]. Extensive evidence is available to support *S. indica* as a biocontrol agent, such as resistance against the black spot of cabbage caused by *Alternaria brassicicola* [3], Stemphylium leaf blight of onion [28], *Rhizoctonia solani* in tomato [113], Verticillium wilt caused by *Verticillium dahliae* [112], and root rot caused by *Fusarium graminearum* [114]. The effect of *S. indica* on biotic stress in various crops is summarised in Table 2.

To develop resistance against various pathogens, *S. indica* builds molecular relations with the host plants in addition to following physiological and biochemical pathways [121]. Many hypotheses have been proposed to explain the use of *S. indica* as a biocontrol agent. As suggested by Li et al. [122], *S. indica* follows indirect and complex mechanisms to trigger pathogen resistance in plants, such as intervention in MAMPs, which include oxidative burst and upregulation of defence-related genes. Moreover, Li et al. [122] reported the involvement of *S. indica* in mitogen-activated protein (MAP) kinase and phenylpropanoid biosynthesis pathways, which are responsible for plant immunity. Therefore, *S. indica* triggers an immune response in the host plant by modifying the transcriptomic changes induced by pathogens. In the case of leaf pathogens, systemic resistance was triggered in tomato, maize, wheat, barley, and *Arabidopsis*. It has been associated with jasmonic acid (JA)-dependent signalling [123], synthesis of antioxidants [77,124], reducing H_2_O_2_ content [99], elevating relative water content, increasing the membrane stability index [122], and upregulating the gene expression in defence-related pathways [115,125]. However, the fungus, induces direct antioxidant action against root pathogens [28]. Moreover, it was established that under stress conditions, *S. indica* activated antioxidant capacity in the host plant by increasing the concentration of antioxidant enzymes, such as ascorbate peroxidase (APX), superoxide dismutase (SOD), glutathione peroxidase (GPX), and catalase (CAT). These enzymes, associated with induced systemic resistance, are of key importance in defence mechanisms. They cause the detoxification of reactive oxygen species generated by pathogens and thus act as a barrier against pathogen attacks [28,69]. Furthermore, Roylawar et al. [28] reported a decrease in malondialdehyde accumulation, resulting in protection against ROS damage. Peroxidase, on the other hand, can also stimulate lignin and phenol biosynthesis, which hinders the growth of the pathogen [126]. In addition, *S. indica* alters the synthesis of defence hormones, such as ethylene, thus protecting the host plant against different pathogens [9].

*S. indica* also mediates the upregulation of defence-related genes, such as the lipoxygenase (*LOX)* gene, leading to increased expression of *AcLOX1* and *AcLOX2* lipoxygenases and *AcCHI* chitinase, which are considered essential for salicylic acid (SA) and jasmonic acid (JA) pathways, thereby regulating induced systemic resistance (ISR) [28,127,128]. It was suggested that the application of *S. indica* upregulated the expression of *WRKY* genes involved in defence pathways, resulting in resistance against early blight disease in tomatoes and sharp eyespot and Fusarium head blight in wheat through modulating the expression of pathogenesis-related (PR) genes [115,122]. Moreover, a recent discovery by Ntana et al. [13] reported an upregulation of the expression of the terpenoid synthase gene (*SiTPS*) upon the colonisation of *S. indica* in tomato roots. It resulted in the production of viridiflorol, which inhibits the growth of the phytopathogenic fungus *Colletotrichum truncatum*. In *S. indica*-colonised taro crop, Lakshmipriya et al. [69] found activation of defence-related genes, such as senescence-associated genes, as well as genes encoding cytochrome P450, Delta (12) oleic acid desaturase FAD2, and calcium-dependent protein kinases (*CDPKs*), resulting in the reduction in taro leaf blight. Figure 6 depicts the effect of *S. indica* on host plants under biotic stress.

Based on these findings, it is reasonable to conclude that *S. indica* has a favourable impact on disease tolerance in a range of plants and might be exploited to be developed into an effective biocontrol agent.

### 4.2. Abiotic Stress Management by S. indica

Crop plants are also exposed to various types of abiotic stresses, such as drought, floods, salinity, heavy metal toxicity, and low-high temperatures, which affect crop development and productivity [129]. *S. indica* has proven to minimise the abiotic stress levels in several crops, as presented in Table 3. The mechanism involves the establishment of a ROS scavenging system [27,51,68,121,130], the activation of genes, such as those encoding dehydration responsive element binding proteins 2A (*DREB2A*) and calcineurin B-like protein (*CBL1*), responsive-to-dehydration 29A (*RD29A*) gene, and other defence-related genes (*PR*, *LOX2,* and *ERF1* genes), as well as the promotion of the production of organic osmolytes, such as proline [14,26] and glycine betaine [14]. However, Abdelaziz et al. [131] reported a decrease in proline content in *S. indica*-treated plants, which might be attributed to an improvement in plant response toward drought stress following the application of *S. indica*. It was further proposed that *S. indica* colonisation not only increased the plant’s hormonal and metabolic activity, but also controlled the ionic homeostasis of sodium (Na^+^) and potassium (K^+^) ions. Therefore, protecting the plants against stress [132]. Figure 7 summarises the impact of *S. indica* on plants under abiotic stress.

#### 4.2.1. Drought Stress

One of the most devastative abiotic stresses, causing significant crop loss is a drought stress which needs to be mitigated. In this context, several studies have demonstrated the positive effects of *S. indica* on various crops under drought stress conditions, including higher water use efficiency (WUE), nutrient uptake, and rise in chlorophyll content by upregulating chlorophyll biosynthesis genes [44,51,131,133]. Additionally, it increased the activities of antioxidant enzymes, such as catalase (CAT) and peroxidase (POD) [101,131,134]. Upregulation of the ROS scavenging system, accumulation of soluble proteins, increases in sugars and amino acids, such as proline [26,27,101,135], as well as changes in hormones, such as an increase in indole-3-acetic acid (IAA) content, a decrease in ABA and ethylene [131], and a reduction in malondialdehyde (MDA) levels [44,131] were also reported.

Furthermore, *S. indica* improves morphological parameters under drought conditions, such as root length, shoot length, grain yield and biomass, thus confirming resistance against drought in several crops, including globe artichoke [136], maize [137], Chinese cabbage [68], walnut [27], rice [44] and sesame [138]. Various drought-related genes, such as responsive-to-dehydration 29A gene *(RD29A)*, (*ANAC072)*, phospholipase D delta (*PLDδ*), and those encoding early response to dehydration 1 *(ERD1),* dehydration-response element binding protein 2A *(DREB2A)*, salt and drought-induced ring finger 1 *(SDIR1)*, calcineurin B-like protein (*CBL1*), CBL-interacting protein kinase 3 *(CIPK3)*, and histone acetyltransferase (*HAT*) were upregulated following the application of *S. indica*, resulting in drought stress tolerance in the *Arabidopsis* plant [39]. Moreover, Azizi et al. [50] reported enhancement in the expression of late embryogenesis abundant 14 (*LEA14*), a dehydrin (*TAS14*) and pyrroline-5-carboxylate synthase (*P5CS*) gene under drought stress after the application of *S. indica* in tomato. Upregulation of these genes is known to ameliorate drought tolerance in plants. In addition to gene regulation, it has been observed that *S. indica* increased the calcium sensing regulator (CAS) protein, which regulates the stomatal movements in Chinese cabbage leaves, thus inducing drought tolerance [68].

#### 4.2.2. Cold Stress

Tolerance to cold stress has been reported in certain studies following the application of *S. indica* to host plants. According to Li et al. [139], banana crops responded positively to *S. indica* colonisation under cold stress due to accumulating osmolytes, activating antioxidant capacity, and increasing the expression of the cold stress-related genes in banana leaves. *S. indica* colonisation has also resulted in cold stress tolerance in *Arabidopsis* by increasing ascorbic acid content, proline content, phytohormone levels, such as brassinolide (BR), abscisic acid (ABA), and regulating cold stress-related genes, such as C-repeat-binding factor (*CBFs*), cold-regulated genes (*CORs*), brassinazole-resistant (*BZR1*), senescence-associated gene (*SAG1*), and pyrabactin resistance 1-like (*PYL6*). Moreover, lower H_2_O_2_ and malondialdehyde (MDA) levels were recorded in *Arabidopsis* seedlings following *S. indica* colonisation [140]. Furthermore, *S. indica* increased the yield of barley crops under low temperatures by enhancing nutrient uptake [42].

#### 4.2.3. Salinity Stress

Salinity stress is detrimental to agricultural production and results in massive crop losses. However, the application of *S. indica* has been reported to promote plant growth under salt stress by enhancing chlorophyll content, biomass, root growth, and antioxidant activity in *Arabidopsis thaliana*, *Cucumis melo*, and *Medicago truncatula* [90,141,142]. Singh et al. [143] reported that *S. indica* increased antioxidant enzymes, carotenoid, α-tocopherol, and proline levels while decreasing relative membrane permeability, lipid peroxidation, and lipoxygenase enzyme activity in wheat under salinity stress. *S. indica* has shown to mitigate the salt stress in other crops by increasing proline content [37]. Furthermore, salt tolerance induced by *S. indica* is mediated by the high osmolarity glycerol (HOG) mitogen-activated protein (MAP) kinase signalling pathway, in which the upregulation of genes, such as *SiHOG1* plays an important role in root colonisation and improves the condition of the host plant under salt stress [144]. In barley crops, *S. indica* governs the ethylene biosynthesis pathway and carbohydrate and nitrogen metabolism, thus resulting in salt tolerance [145]. Moreover, salinity stress was ameliorated in Chinese cabbage by modulation of phytohormones, such as salicylic acid and gibberellic acid, as well as by enhancing the activities of antioxidant enzymes [146]. Furthermore, this endophytic fungus conferred resistance to salt stress in tomato seedlings owing to its ability to boost the uptake of nutrients, such as nitrogen, phosphorus, and calcium, and to enhance K^+^/Na^+^ homoeostasis by regulating the expression of various genes, such as sodium/hydrogen exchanger 2 (*NHX2*), Na^+^/H^+^ antiporter (*SOS1*), and cyclic nucleotide-gated channel (*CNGC15*). These genes regulate the function of aquaporins, thus maintaining water status in plants [147]. Additionally, Boorboori et al. [148] recently highlighted the relevance of *S. indica* in resisting drought and salt stress in plants, consequently increasing agricultural output.

#### 4.2.4. Heavy Metal Stress

Alleviation of heavy metal stress by application of *S. indicia* has been reported in several host plants by reducing the concentration and absorption of heavy metals, such as lead, arsenic, and cadmium in various plants, such as sunflower, wheat, barley, and alfalfa [41,52,149,150,151]. Moreover, the concentration of lead was alleviated in the shoots of the sweet basil plant with the application of *S. indica* [152]. This effect of reducing the concentration of heavy metals by *S. indica* colonisation can be attributed to the binding of these heavy metals to the cell wall of fungal hyphae, thereby reducing their uptake into the aerial plant parts [60]. Mohd et al. [153] suggested that the compartmentalisation, adsorption, and precipitation of these metals are thought to limit their mobility, resulting in decreased translocation into the shoot.

Additionally, a significant improvement in growth parameters has been recorded in host plants treated with *S. indica* under metal toxicity conditions, such as an increase in photosynthetic rate and pigments, biomass, antioxidant enzymes, flavonoids, and proline content, and a reduction in H_2_O_2_ and MDA content in the host plant, including rice [108], wheat [150], sunflower [149], and tobacco [154]. Moreover, an increase in the expression of stress-related genes, such as phytochelatin synthase-related genes (*TaPCS1, oas1, Gsh2*), *GPX,* and heat shock proteins (*Hsp70*) was reported in tobacco under cadmium stress in *S. indica* colonised plants [154]. Furthermore, *S. indica* can act as a detoxifying agent in heavy metal-contaminated soils by immobilising these metals in the root zone, limiting their release, and improving soil quality [80].

**Table 3 plants-11-03417-t003:** Effects of *S. indica* on host plants under abiotic stress conditions.

Host Plant	Abiotic Stresses	Effect of *S. indica* on Host Plant under Abiotic Stress Conditions		Reference
Eggplant (*Solanum melongena*)	Drought	Increase in root and shoot length, biomass, proline, chlorophyll, and water content, catalase activity, and guaiacol peroxidase activity	-	[155]
Chinese cabbage (*Brassica oleracea*)	Drought	Increase in chlorophyll content, antioxidant enzyme activity, drought-related genes expression, and CAS protein	-	[68]
Globe artichoke (*Cynara scolymus*)	Drought	Increase in plant biomass, photosynthetic pigment, carotenoid, proline, and potassium levels in leaves	Decrease in sodium content	[136]
Tomato (*Lycopersicon* esculentum)	Drought	Increase in shoot fresh and dry weight, proline and relative water content	-	[50]
Rice (*Oryza sativa*)	Drought	Increase in grain yield, stomatal closure, leaf surface temperature, antioxidant enzyme activity	-	[44]
		Increase in seedling growth, seedling biomass, uptake of phosphorus and zinc, chlorophyll fluorescence, proline, and total antioxidant capacity in leaves	-	[156]
Maize (*Zea mays*)	Drought and mechanical stress	Increase in plant adaptability, root length and volume, relative water content, leaf water potential, and chlorophyll content	-	[26]
Maize *(Zea mays)*	Drought	Increase in leaf size, root length, and number of tap roots	-	[101]
		Increase in oxidative potential of the roots, and expression of genes responsible for hormonal functions	-	[107]
Barley *(Hor deumvulgare*)	Drought	Increase in fresh and dry weight of plant, root growth, number of tillers, proteins involved in photosynthesis, and antioxidant enzyme activity	-	[157]
		Increase in activity of photosystem and electron transport chain and accumulation of drought responsive proteins	-	[135]
Finger millets (*Eleusine coracana*)	Drought	Increase in plant growth, biomass, total chlorophyll, proline, total and soluble sugar content	Decrease in MDA and hydrogen peroxide content	[158]
Walnut (*Juglans regia*)	Drought	Increase in plant height, total fresh biomass, root/shoot ratio, relative growth rate, leaf relative water and chlorophyll content, gas exchange parameters, photochemical efficiency, photochemical quenching, and photosystem II quantum yield, osmotic adjustment and antioxidant activity	-	[27]
Quinoa (*Chenopodium quinoa*)	Drought	Increase in total biomass, water balance, and leaf gas exchange	-	[51]
Aloe vera (*Aloe barbadensis*)	Salinity	Increase in plant biomass, shoot and root length, number of shoots and roots, chlorophyll, gel, phenols, flavonoid, and aloin content, and radical scavenging activity	-	[78]
Gerbera *(Gerbera jamesonii*)	Salinity	Increase in plant biomass, photosynthesis, and salt tolerance	Decrease in MDA and hydrogen peroxide content	[86]
Melon *(Cucumis melo*)	Salinity	Increase in antioxidant content, dry weight	Decrease in salinity stress	[90]
Sweet basil *(Ocimum basilicum*)	Salinity	Increase in plant dry weight area, essential oil content, and yield	-	[85]
Tomato (*Lycopersicon esculentum)*	Salinity	Increase in plant growth, photosynthetic pigments, proline and glycine betaine, potassium content, water potential, net photosynthesis, stomatal conductance, and transpiration rate	-	[147]
Arabidopsis (*Arabidopsis thaliana*)	Salinity	-	Decrease in sodium content	[159]
Rice (*Oryza sativa)*	Salinity	Increase in fresh weight, dry weight, length of root and shoot, photosynthetic pigment, and salt tolerance	-	[37]
		Increase in proline concentration and antioxidant enzyme activity, potassium concentration, salt tolerance	Decrease in MDA and sodium ion concentration	[160]
Barley (*Hordeum vulgare)*	Salinity	Increase in plant growth, ascorbic acid activity of antioxidant enzymes, metabolic heat efflux, and fatty acid desaturation in leaves	Decrease in NaCl-induced lipid peroxidation	[161]
Banana (*Musa acumunata*)	Cold	Increase in superoxide dismutase (SOD) and catalase (CAT) activity and soluble sugar (SS) and proline content	Decrease in MDA and hydrogen peroxide content	[139]
Barley (*Hordeum* vulgare)	Cold	Increase in grain dry weight, nutrient uptake, and early flowering	-	[42]
Arabidopsis (*Arabidopsis thaliana*)	Cold	Increase in soluble proteins, proline and ascorbic acid, cold tolerance	-	[140]
Banana (*Musa acumunata*)	High temperature	Increase in proline content, MDA, hydrogen peroxide and ABA, and resistance to high temperature	Decrease in IAA and GA content in banana leaves	[162]
Heavy metal stress				
Artemisia (*Artemisia annum*)	Arsenic	Increase in plant growth, biomass, flavonoids, artemisinin, super oxidase dismutase, peroxidase activity, phenolic acid, and phenolic compounds	-	[80]
Wheat (*Triticum* aestivum)	Cadmium	Increase in plant growth, photosynthetic pigments, antioxidant enzymes, proline content	Decrease in cadmium accumulation in stem and root	[150]
Rice (*Oryza sativa*)	Arsenic	Increase in plant growth, chlorophyll content	Decrease in MDA content and arsenic uptake	[108]
	Cadmium	Increase in nutrient uptake, photosynthesis, antioxidant enzyme activity, maintenance of cellular structures	Decrease in cadmium concentration in shoot	[163]
Barley (*Hordeum vulgare)*	Arsenic	Increase in iron concentration	Decrease in arsenic concentration in shoot	[151]
Sunflower (*Helianthus annuus)*	Cadmium	Increase in plant growth, chlorophyll content, proline content, Fv/Fm, and electron transport rate values	Decrease in cadmium content in plant	[149]
Alfalfa (*Medicago sativa)*	Cadmium	Increase in growth rate, shoot dry weight, antioxidant enzyme activity	Decrease in cadmium concentration in shoot	[84]
Tobacco (*Nicotiana* tabacum)	Cadmium	Increase in chlorophyll content, antioxidant enzymes, and proline content, enhanced expression of stress-related phytochelatin biosynthesis genes	Decrease in MDA	[154]
Soybean *(Glycine max*)	Cadmium	Increase in plant growth, dry weight, shoot height and photosynthesis, transpiration ratio and stomatal conductance, and antioxidant activity	Decrease in cadmium content	[164]

According to research on *S. indica*, this endophytic fungus could serve as a potential biofertiliser, reducing various biotic and abiotic challenges while enhancing plant development and, eventually, agricultural productivity.

## 5. Interaction of *Serendipita indica* with Nanoparticles

Nanotechnology, an emerging science, is gaining immense attention in the field of agriculture due to its small size and higher efficiency over bulk materials [165,166]. Therefore, integrating this technology with beneficial microorganisms could unfold novel approaches for improving agricultural outputs. It was reported that applying *S. indica* and copper nanoparticles together increased plant growth, making the combination useful as a nanofertiliser [165]. Varma et al. [167] developed nano-embedded *S. indica*, as a potential biofertiliser, including a combination of *S. indica* and nanoparticles, such as zinc oxide, titanium oxide, and carbon nanotubes. The interaction between nanoparticles and the fungus resulted in increased fungal biomass, spore number, thicker hyphae, and fewer vacuoles [167]. Rane et al. [168] suggested the combined application of calcium phosphate nanoparticles with *S. indica* to be a more efficient plant growth enhancer than a sole application of nanoparticles or endophytic fungus. Another study revealed that the combined application of nitrogen nanoparticles with *S. indica* did not show significant improvement in the treated plants, but the application of *S. indica* alone was found to be better than nanoparticles [3]. However, insufficient understanding of the usage and application of nanoparticles in agriculture might lead to significant risks to humans and the environment. These particles can penetrate deep into the biological system, resulting in serious threats to both mammals and the environment [169]. Moreover, organic nanoparticles produced biologically pose less risk than those produced using chemical and physical methods [170]. Nevertheless, each technology has its pros and cons; therefore, the vital factor is using the optimum dosage and appropriate techniques for applying nanotechnology in the agricultural sector to control the risks associated with nanoparticles [171].

As a result, additional research on the beneficial application of nanomaterials and plant growth-promoting microorganisms is required to develop more efficient and practical nano biofertilisers for agricultural production.

## 6. Conclusions

Based on the above discussion, we presented the influence of the beneficial endophytic fungus on plant growth, quality, and yield, thus improving global food security and the nutritional prospects of the produce. *S. indica* ensures environmental protection while being economical at the same time. In addition, the protection of plants from biotic and abiotic stresses is attained through the application of *S. indica*. Moreover, its wide range of host acceptability and its axenic culturing ability have made it one of the promising beneficial microorganisms in the field of agriculture. However, a gap still exists between the research and its widespread release, which needs to be bridged by focusing on the nutrient uptake mechanisms mediated by *S. indica* and conducting in-depth research on molecular mechanisms that would create opportunities for the commercial viability of endophytic fungus *S. indica.*

## Figures and Tables

**Figure 1 plants-11-03417-f001:**
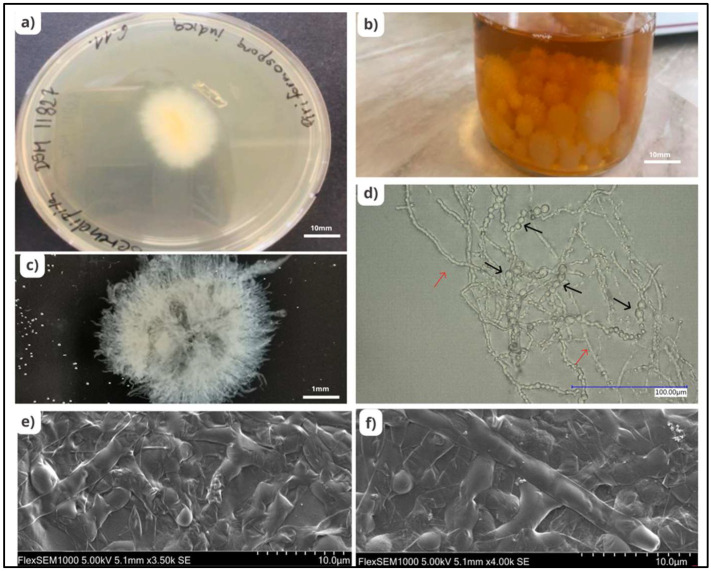
Morphology of *Serendipita indica*. (**a**) Growth of *S. indica* colony on PDA media; scale bars of 10 mm. (**b**) Growth of *S. indica* colony in PDB media; scale bars of 10 mm. (**c**) Growth of *S. indica* from PDB medium; scale bars of 1 mm. (**d**) Chlamydospores (black arrows) and mycelium (red arrows) of *S. indica* observed under a bright field microscope; scale bars of 100 µm. (**e**,**f**) Mycelial structure of *S. indica* observed under an electron microscope.

**Figure 2 plants-11-03417-f002:**
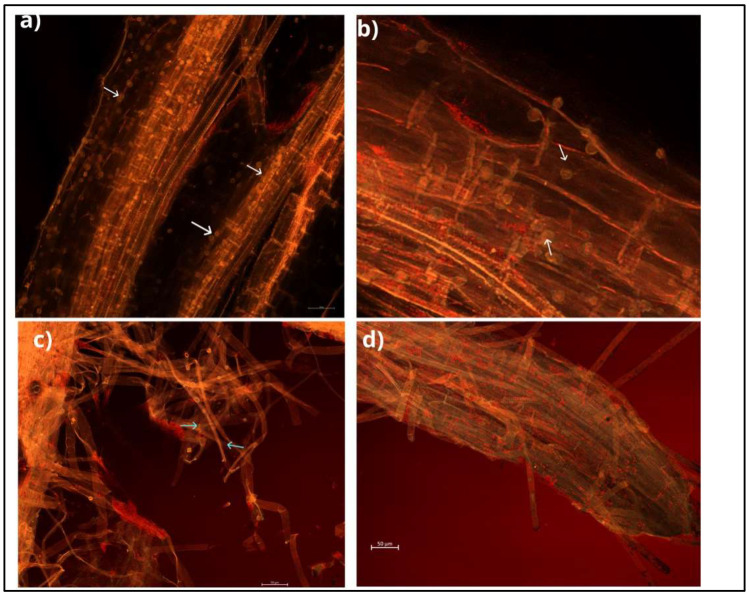
The roots of cabbage inoculated with *S. indica* observed under a fluorescent microscope; scale bars of 50 μm. (**a**) The cortex of cabbage roots covered with spores. (**b**) Close view of spores covering cabbage roots. (**c**) The fungal mycelium developing externally. (**d**) The fungus covering the root tip. White and blue arrows point to spores and mycelium, respectively.

**Figure 3 plants-11-03417-f003:**
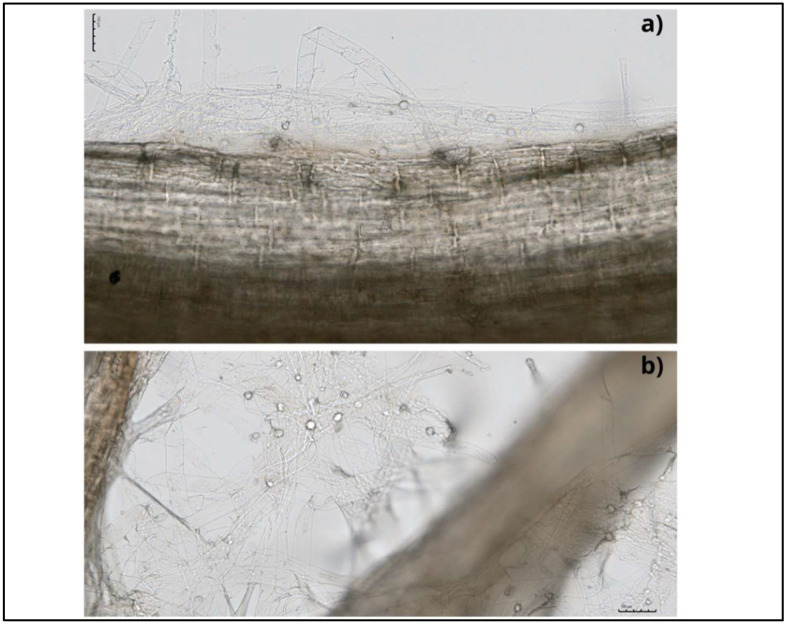
Bright-field microscope photos of the cabbage roots with internal (**a**) and external (**b**) colonisation by *S. indica*; scale bars of 100 µm.

**Figure 4 plants-11-03417-f004:**
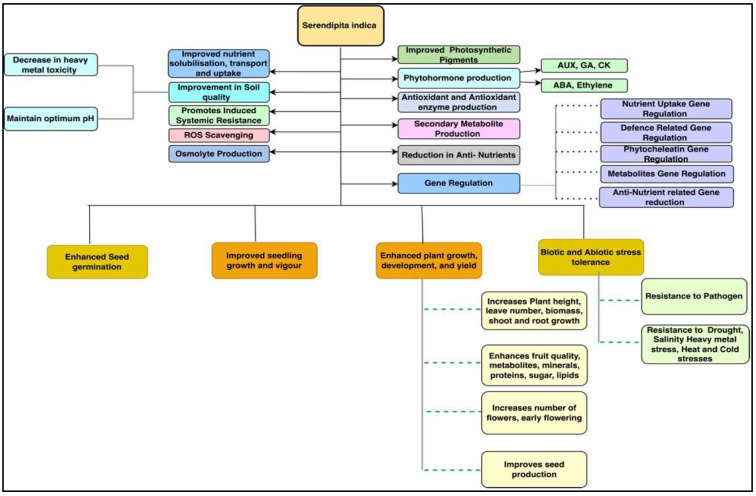
Schematic representation of the effects of *S. indica* on plant growth, development, and response to various stresses.

**Figure 5 plants-11-03417-f005:**
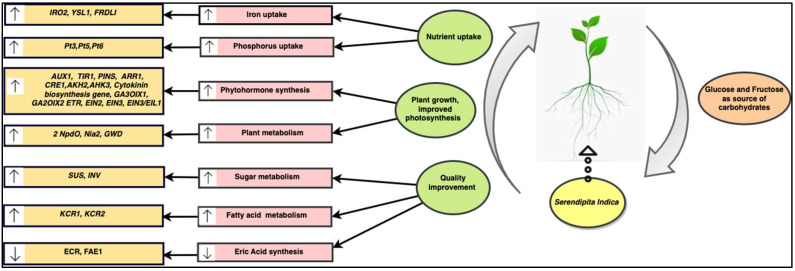
Mutual interaction of *S. indica* with host plant resulting in enhanced plant metabolism by regulating the plant growth mechanism at molecular level. Genes which exhibit a fundamental role in this process are *AUX1*, *PINS*, *TIR*—auxin biosynthesis genes, *AHK2*, *AHK3*, *CRE1*—cytokinin receptor genes, *ARR*—cytokinin-responsive gene, *ECR*—trans-2, 3-enoyl-CoA reductase, *ETR*, *EIN2, EIN3*, *EIN3/EIL1*—ethylene biosynthesis genes, *FAE1*—3-ketoacyl-CoA synthase, *FRDL1*, *IRO2, YSL1*—iron transporter genes, *GA3OIX1, GA2OIX2*—gibberellic acid biosynthesis genes, *GWD*—glucan-water dikinase, *INV*—invertase, *kcr1*—3-ketoacyl-CoA reductase 1, *kcr2*—3-ketoacyl-CoA reductase 2, *2NpdO*—nitropropane dioxygenase, *Nia2*—nitrate reductase, *PT3*, *PT5,* and *PT6*—phosphorus transporter genes. Upward and downward arrows represent upregulation and downregulation of genes or processes, respectively.

**Figure 6 plants-11-03417-f006:**
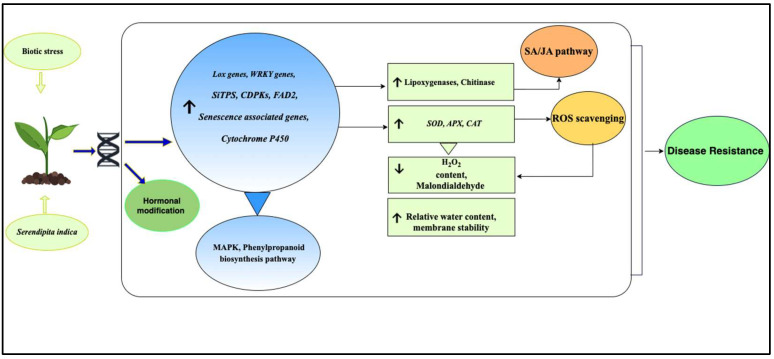
Schematic representation of the effects of *S. indica* on plants under biotic stresses. *S. indica* maintains the plant growth and development throughout the growing stages by regulating various genes and antioxidant production, leading to the development of tolerance against stresses in host plants. Various genes are involved in the process consisting of *FAD2*—fatty acid desaturase 2, *LOX*—lipoxygenase, *SiTPs*—terpenoid synthase gene, senescence associated genes, *WRKY*—transcription factor. Upward and downward arrows represent upregulation and downregulation of genes or processes, respectively.

**Figure 7 plants-11-03417-f007:**
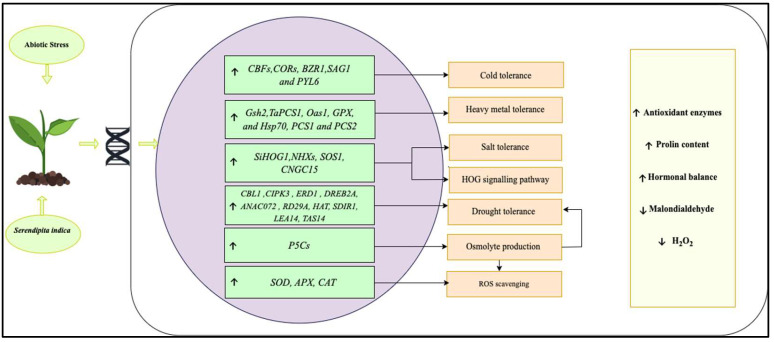
Schematic description of the effects and the molecular mechanisms of *S. indica* on host plants under various abiotic stress. Cold tolerance genes: *BZR1*—brassinazole-resistant, *CBFs*—C-repeat-binding factor, *CORs*—cold-regulated genes, *PYL6*—pyrabactin resistance 1-like, *SAG1*s—senescence-associated gene; Drought tolerance genes: *ANAC07*—transcription factor gene, *CBL1*—calcineurin B-like protein, *CIPK3*—CBL-interacting protein kinase 3, *DREB2A*—dehydration-response element binding protein 2A, *ERD1*—early response to dehydration 1, *HAT*—histone acetyltransferase, *LEA14*—late embryogenesis abundant 14, *P5CS*—pyrroline-5-carboxylate synthase, *RD29A*—responsive-to-dehydration 29A gene, *SDIR1*—salt and drought-induced ring finger 1, *TAS14*—a dehydrin; Heavy metal tolerance genes: *Gsh2, PCS1, PCS2, oas1, TaPCS1*—phytochelatin synthase-related genes, *GPX*—glutathione peroxidase, *Hsp70*—heat shock proteins; Salt tolerance: *CNGC15*—cyclic nucleotide-gated channel, *NHX2*—sodium/hydrogen exchanger 2, *SiHOG1*—*S. indica* high osmolarity glycerol, *SOS1*—Na^+^/H^+^ antiporter. Upward and downward arrows represent upregulation and downregulation of genes or process, respectively.

**Table 1 plants-11-03417-t001:** Effects of *S. indica* on growth and quality of host plants.

Host Plant	Method of Application	Effect of *S. indica* on Growth	Effect of *S. indica* on Quality	Reference
Tomato (*Lycopersicon esculentum*)	Seedling root application	Increase in plant growth, yield, early flowering	Improved firmness in fruits, total soluble solids, and acidity	[49]
Lettuce (*Lactuca sativa*)	Seedling application	Increase in plant height, fresh and dry weight, yield	Increase in chlorophyll, nitrogen, phosphorus, and potassium content	[67]
	Seedling root application		Increase in zinc and manganese content	[64]
Chinese cabbage (*Brassica oleracea*)	Seedling root application	Increase in root and shoot growth, biomass, and lateral root formation	_	[68]
Cabbage (*Brassica oleracea)*	Substrate application and Seed application	Increase in plant height, number of leaves, NDVI, fluorescence level (Ft)	Increase in chlorophyll and carotenoids content, antioxidant capacity	[3]
Bell pepper (*Capsicum annuum*)	Seed treatment	Increase in plant growth and yield	_	[19]
Spinach (*Spinacia oleracea*)	Seed application	Increase in plant height, dry and fresh weight	_	[11]
Taro (*Colocasia esculenta*)	Sterile soil application	Increase in plant height, number and area of leaves	Increase in total phenol content and defence-related enzymes	[69]
Sweet potato (*Ipomea batatas*)	Seedling irrigation	Increase in plant biomass, number of leaves, and lateral roots	Increase in photosynthetic pigments, catalase, JA-mediated activity	[47]
Wheat (*Triticum aestivum*)	Seedling roots	Increase in shoot dry biomass	Increase in nitrogen, phosphorus, and iron uptake	[52]
Rice (*Oryza sativa*)	Seed and Seedling roots	Increase in plant growth, biomass, yield	NPK, chlorophyll, and sugar content	[59]
	Seed treatment and Soil treatment	Increase in dry weight of plant	Increase in phosphorus and potassium uptake	[46]
Barley (*Hordeum vulgare*)	Seed application	Increase in crop yield	_	[70]
Finger Millets (*Eleusine coracana*)	Seed application	Increase in plant height, number of tillers, plant biomass, ear heads, test weight, grain, and dry straw yield	Increase in NPK content	[71]
Groundnut (*Arachis hypogaea*)	Seedling application	Increase in growth and number of pods, seeds per plant, shelling percentage, 100-seed weight, and pod yield	_	[72]
Green gram (*Vigna radiata*)	Seed application	Increase in number of nodules per plant, leaf area, and yield	Increase in minerals uptake, chlorophyll content	[73]
Chickpeas (*Cicer arietinum)*	Seed application	Increase in growth and yield	Increase in phosphorus (P) uptake	[74]
Sugarcane (*Saccharum* sp.)	Plantlets	Increase in growth and yield, cane height, tillering	Increase in iron and copper content, sugar content	[66]
Rapeseed (*Brassica napus*)	Seedling application	Increase in plant yield, biomass, early flowering	Increase in oil content, nutrient uptake, decrease in anti-nutrient content	[25]
Black pepper (*Piper nigrum*)	Root cuttings	Increase in fresh and dry weight, number and area of leaves, early flowering	Increase in chlorophyll content	[43]
Sunflower (*Helianthus*)	Seedling application	Increase in growth of plant and seed yield.Increase in root and stem growth, number and area of leaves. Increase in flower diameter, dry weight, and total biomass	Increase in seed oil content	[40]
Turmeric (*Curcuma* sp.)	Bud application	Increase in productivity	Increase in secondary metabolite curcumin and oil content	[75]
Fennel (*Foeniculum vulgare*)	Seedling	Increase in plant height, dry weight of plant, fruit dry weight	Increase in essential oil content (anethole)	[45]
Thyme (*Thymus vulgaris*)	Shoot application	Increase in plant height, fresh and dry weight of shoot. Increase in root length, fresh and dry weight of roots	Increase in essential oil content (thymol)	[76]
Bacopa (*Bacopa monnieri*)	Micro propagated plants	Increase in plant growth	Increase in bacoside and antioxidant content	[77]
Aloe vera (*Aloe barbadensis*)	Root application	Increase in biomass, shoot and root length, shoot and root number	Increase in chlorophyll, gel, aloin, and phenol content in leaves	[78]
Artemisia (*Artemisia annua*)	Seedling root application	Increase in plant height, dry weight, leaf yield	Increase in chlorophyll, phosphorus, nitrogen, flavonoids, and artemisinin content	[79]
	Root application			[80]
*Centella asiatica*	Root application	Increase in shoot and root biomass	Increase in asiaticoside content	[81]
*Coleus forskohlii*	Root application	Increase in aerial biomass, flower development	Increase in chlorophyll content, phosphorus acquisition	[82]
*Lantana camara*	Suspension culture of plant	_	Increase in triterpenoids (ursolic acid, oleanolic acid, and betulinic acid) production	[83]
Pine vines (*Aristolochia elegans* Mart.)	Substrate application	Increase in plant height, number and length of leaves, total biomass	Increase in aristolochic acid in leaves	[30]
Alfalafa (*Medicago sativa*)	Seedling application	Increase in biomass, shoot dry weight	Increase in nutrient uptake, antioxidant enzyme activity	[84]
Sweet basil (*Ocimum basilicum*)	_	Increase in plant growth, leaf area, leaf dry weight, yield	Increase in oil content (Geranial, Neral, and Estragole)	[85]
Gerbera (*Gerbera jamesonii*)	Root seedling application	Increase in above and underground plant biomass	Increase in photochemical efficiency	[86]
*Anthurium* sp.	Seedling roots	Increase in plant and root growth	Increase in phosphorus uptake and chlorophyll content	[87]
*Lolium multiflorum*	Seed application	Increase in plant height, basal diameter, biomass relative growth rate	Increase in leaf relative water content and chlorophyll content	[38]
Banana (*Musa acuminata*)	Plantlet roots	Increase in plant height, number and area of leaves, stem diameter, number of suckers, number of fingers per bunch, and yield per plant	Increase in chlorophyll, nitrogen, and phosphorus content	[88]
Passion fruit (*Passiflora edulis*)	Soil application	Plant growth in later stages, fruit size	Increase in fruit quality and secondary metabolites	[89]
Trifolium orange(*Poncirus trifoliata*)	Substrate application	Increase in plant height, number of leaves, leaf, stem, and root biomass	Increase in nitrogen, phosphorus, and magnesium content	[7]
Melon *(Cucumis melo*)	Substrate application	Increase in fresh and dry weight of plants	Increase in chlorophyll content	[90]
Pineapple (*Ananas comosus*)	Substrate inoculation in root zone	Increase in plant height, number of leaves, and shoot dry weight	Increase in photosynthetic efficiency, nitrogen, phosphorus, potassium, calcium, and magnesium content	[61]

**Table 2 plants-11-03417-t002:** Effects of *S. indica* on host plants under biotic stress conditions.

Host Plant	Biotic Stress	Effect of *S. indica* under Biotic Stress		Reference
Tomato *(Lycopersicon esculentum)*	Verticillium wilt	Increase in plant growth, dry and fresh weight content, disease resistance	—	[112]
	Early blight	-	Decrease in disease severity	[115]
		Increase in plant growth, systemic defence response	-	[116]
Tomato *(Lycopersicon esculentum)*	Leaf curl virus	Increase in root and plant growth, quality, and yield of fruit	Decrease in disease incidence	[49]
Chinese cabbage (*Brassica oleracea*)	Club root	Increase in biomass, flavonoids	Decrease in gall formation	[117]
Cabbage (*Brassica oleracea*)	Black spot	Increase in plant growth, chlorophyll content, antioxidant capacity, nitrogen content	Decrease in disease severity	[3]
Onion (*Allium cepa)*	Stemphylium Leaf Blight	Increase in leaf growth and root biomass, enzymatic activity, defence-related genes	Decrease in disease severity	[28]
Taro (*Colocasia esculenta*)	Leaf blight	Increase in plant growth, chitinase, β-1, 3 glucanase and total phenol, defence-related genes, disease resistance	-	[69]
Soyabean (*Glycine max)*	Cyst nematode	Increase in plant growth and development	Decrease in nematode egg density	[118]
Chickpea (*Cicer arietinum*)	Grey mold	Increase in biomass, root growth, antioxidant enzyme defence system, protection from pathogen	-	[63]
Anthurium (*Anthurium*)	Bacterial wilt	Increase in biomass, plant and root growth, photosynthetic rate, phosphorus uptake, disease resistance	-	[87]
Rhododendron (*Rhododendron catawbiense*)	Dieback	Delayed disease	-	[119]
Blueberry (*Vaccinium corymbosum)*	Dieback	Increase in plant growth	Decrease in disease severity and mortality rate	[120]

## Data Availability

All data generated or analysed during this study are included in this published article.

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
