# Peer review of "Serendipita indica—A Review from Agricultural Point of View"

_plants, 2022, doi:10.3390/plants11243417_

Round 1

Reviewer 1 Report

The topic of this manuscript is very interesting and the manuscript is well organized. However, there are so many formatting and punctuation errors and language problems. Therefore, I recommend major revisions. Detailed comments were listed as follows.

1. 'metal moment'?

2. 'These microorganisms include bacteria, fungi, actinomycetes, symbiotic or nonsymbiotic microbes, mycorrhizal and endophytic fungi'--fungi include mycorrhizal and endophytic fungi, please correct this sentence.

3. This fungus was once named as Piriformospora indica, please describe about this.

4. Figure 1a: Piriformospora indica?

5. Figure 1c: fungus?

6. Reference citation in the figure legends are necessary.

7. 'Rokni et al. [19] also demonstrated that optimization of inoculum quantity of S. indica affects the growth of the host plant which was improved by applying 1-3% w/w of S. indica as a seed treatment. Higher concentration however, did not report any beneficial effects. Besides this production of calcium and lectin protein kinase are suggested to be the early signals of the S. indica colonisation with the host plant [20], [21].' --these sentences are hard to understand, please rewrite.

8. 'ability to manage various biotic and abiotic stresses' ?  did you mean 'resist'?

9.' quality of produce'?

10. all gene names should be in italic. And S. indica should be always in italic.

11. the citation format of references in the main body should be corrected. For example, '[21], [22]' should be corrected to '[21,22]'.

12: Ghorbani et al. [108], Yang et al.[7]??

13. The language of the whole manuscript should be polished.

Reviewer 2 Report

I recommend the article for publication with minor modifications. There are many typos in the text, and I have color-coded them in the attached file.

Figures 1e and 1f need improvement; they probably do not show what is indicated in the figure description.

Gene names should be italicized.

In Chapter 6, the risks associated with using nanomaterials on mammals, water resources, and the environment should be added. Nanomaterials pose significant risks, and the review should comment on them if it is to be comprehensive.

Round 2

Reviewer 1 Report

The authors improved the quality of this manuscript after corrections. But its language problem is not well solved. And some new errors were made during last round of revision.  Please correct carefully the language problems of this review. I am sure that this manuscript will be very attractive to researchers working on Serendipita indica and growth-promoting microorganisms. 

Besides the language problems, the authors should be noticed that if the figures used in this manuscript have been used in their previously published papers, citations in the figure legends are still very necessary.

Round 3

Reviewer 1 Report

The authors improved greatly the quality of the manuscript. I think the paper can be accepted and published in Plants now.

Author Response

Thank you for your consideration.
